# SympCam: Remote Optical Measurement of Sympathetic Arousal

Björn Braun[1]    Daniel McDuff[2]    Tadas Baltrusaitis[3]    Paul Streli[1]    Max Moebus[1]    Christian Holz[1]

[1]ETH Zurich    [2]University of Washington    [3]Microsoft

{bjoern.braun, paul.streli, max.moebus, christian.holz}@inf.ethz.ch,
dmcduff@uw.edu, tadas.baltrusaitis@microsoft.com

*Abstract*—Recent work has shown that a person's sympathetic arousal can be estimated from facial videos alone using basic signal processing. This opens up new possibilities in the field of telehealth and stress management, providing a non-invasive method to measure stress only using a regular RGB camera. In this paper, we present SympCam, a new 3D convolutional architecture tailored to the task of remote sympathetic arousal prediction. Our model incorporates a temporal attention module (TAM) to enhance the temporal coherence of our sequential data processing capabilities. The predictions from our method improve accuracy metrics of sympathetic arousal in prior work by 48% to a mean correlation of 0.77. We additionally compare our method with common remote photoplethysmography (rPPG) networks and show that they alone cannot accurately predict sympathetic arousal "out-of-the-box". Furthermore, we show that the sympathetic arousal predicted by our method allows detecting physical stress with a balanced accuracy of 90%—an improvement of 61% compared to the rPPG method commonly used in related work, demonstrating the limitations of using rPPG alone. Finally, we contribute a dataset designed explicitly for the task of remote sympathetic arousal prediction. Our dataset contains synchronized face and hand videos of 20 participants from two cameras synchronized with electrodermal activity (EDA) and photoplethysmography (PPG) measurements. We will make this dataset available to the community and use it to evaluate the methods in this paper. To the best of our knowledge, this is the first dataset available to other researchers designed for remote sympathetic arousal prediction.

*Index Terms*—digital health, physiological computing

## I. Introduction

**W**EARABLE sensors, such as smart watches, continue to impact healthcare as they enable people to continuously and non-invasively measure physiological signals where they could not before. These sensors provide valuable data about an individual's health [1]–[4] and can serve to help detect cardiovascular disorders [5], stress [6] or pain [7]. While such wearable sensors have improved substantially, they have some limitations. They have to be worn on the body, are not easily scaled to an entire population, and usually only measure from one location on the body (e.g., the wrist) [8].

In contrast, cameras are versatile sensors that can unobtrusively capture spatial and temporal information at a distance. In addition, most of today's computers and mobile devices are equipped with user-facing cameras for the purpose of video telephony. These properties make cameras attractive as a means to measure physiological signals [9]. Examples

This work was partly funded by the Microsoft Swiss Joint Research Center.

of applications include remote patient monitoring [10] or stress measurement [11]. While, to date, some cardiac and pulmonary signals can be measured using a camera, including heart rate (HR) [12], [13], there are many signals for which there is little or no evidence that cameras alone are sufficient.

One important example is sympathetic arousal, which is a measure of the activation of the sympathetic nervous system. Following many different types of physical, mental, and/or emotional stressors, the sympathetic branch of the autonomic nervous system (ANS) is activated, leading to sympathetic arousal and "fight-or-flight" responses such as increased sweat responses [14]. Sympathetic arousal is, therefore, usually captured with the help of the electrodermal activity (EDA), which measures skin conductivity using electrodes placed on two different locations on the body. Multiple previous works have shown that changes in EDA are a reliable indicator of stress [6], [15] and pain [16], [17]. Traditionally, EDA has been measured at sites on the human body with a high density of sweat glands, such as the fingers or palms, using electrodes that are in steady contact with the person's skin [18]. Most recently, first works have shown that it is also possible to measure EDA and sympathetic arousal completely remotely from videos of the palm and the face [19], [20]. Bhamborea et al. [19] have directly measured EDA by counting the number of specular reflections on the palm. Braun et al. [20] were the first to infer sympathetic arousal from both the face and the palm by measuring blood perfusion, which they have shown correlates with contact EDA. Our goal was to advance the existing approaches and to provide the community with a dataset specifically designed for this task.

In this paper, we present a novel approach for measuring sympathetic arousal from facial videos leveraging neural networks for this task for the first time. Our method extends a 3D convolutional architecture with a temporal attention module (TAM) to learn spatial and temporal-domain features that lead to predictions that highly correlate with gold-standard contact EDA measurements. Our main contributions are:

- A 3D convolutional neural architecture that we tailored to the task of remote sympathetic arousal prediction by introducing a TAM and adapting the temporal dimension to the dynamics of the EDA signal. Using leave-one-subject-out (LOSO) cross-validation, we obtain a mean Spearman correlation of 0.77 between our predicted sympathetic arousal and the ground truth EDA signal, which is an

improvement of 48% compared to previous work [20].

- A dataset with 20 participants on which we evaluate our model. It consists of videos of the face and hand synchronized with measurements of the EDA and PPG signals. We specifically designed the dataset for the task of remote sympathetic arousal assessment and now make it available on request to other researchers because we believe that it opens up new possibilities in the field of telehealth. To the best of our knowledge, this dataset is the first dataset available to other researchers designed to predict sympathetic arousal remotely.

- A classification model that leverages our predicted sympathetic arousal and a remote photoplethysmography (rPPG) signal for the task of detecting physical stress due to pain. We show on this task that our method outperforms rPPG-based approaches by 61%, achieving a balanced accuracy/F1 score of 0.9/0.83 in predicting whether a person is experiencing physical stress due to pain. We highlight the limitations of relying solely on the blood volume pulse (BVP) for detecting physical stress.

## II. RELATED WORK

Compared to self-reports, which are commonly used for stress detection, physiological sensing provides the opportunity for continuous temporal measurements. Traditionally, wearable sensors such as smartwatches were used for physiological measurement. Recently, non-contact (remote) methods, which only use a regular RGB camera, have gained popularity due to their potential for scalability and comfortability [9]. To date, a vast majority of the work in the field of remote physiological sensing has focused on measuring cardiopulmonary signals such as the BVP or the breathing rate. The BVP is inferred from the rPPG signal, which is calculated by measuring the peripheral blood flow via light reflected from the skin [21]–[24]. However, recent work has shown that the HR and other extracted features from the BVP, such as heart rate variability (HRV), are influenced by both sympathetic and parasympathetic activity [25] and, therefore, give only limited information about a person's sympathetic activity [26], [27]. EDA, on the other hand, which measures a person's sweat response, is considered a direct marker of sympathetic activity and is commonly used for psychophysiological evaluations [28], [29].

Previous work using minimally invasive methods has shown that repeated arousal stimuli induced by electrical stimulation are followed by an increase in sympathetic nerve activity, blood flow, and EDA in the forehead [30]. Other work obtained similar results and found that facial blood flow changes due to pain are not dependent on regional (orofacial) stimulation to occure [31]. Furthermore, analysis of thermal imagery found that arousal-induced sweat responses can be detected without contact with the body using thermal cameras [32]. However, thermal cameras are not widely available. Bhamborea et al. [19] published early proof-of-concept results that EDA could be inferred from the palm using only an RGB camera by counting the specular reflections from the skin. Building

on that work [19] and the correlation between blood flow and EDA on the forehead [30], [31], subsequent work has shown that sympathetic arousal can also be inferred from videos of the face using only a regular RGB camera by measuring the peripheral blood flow to the forehead [20]. While this work showed first proof that it is possible to remotely extract a person's sympathetic arousal from a video of the face, the mean correlations across participants were moderate, and the standard deviation (STD) was high. Nevertheless, we were inspired by these results and aimed to develop a more robust method and release a dataset for remote sympathetic arousal prediction that is also available to other researchers. As this previous work [20] indicates that they measure changes in blood flow that correlate with EDA, we will refer to remotely measuring sympathetic arousal instead of measuring EDA.

For camera-based physiological measurements, such as rPPG, supervised neural architectures are state-of-the-art [22], [23], [33]. The spatial information to predict sympathetic arousal from the face should be similar to the spatial information for rPPG prediction. However, the typical frequency band of the sympathetic component of the EDA signal is between 0.045–0.25 Hz [34], which is considerably lower than the frequency band of 0.7–2.5 Hz from the HR [33], [35]. Therefore, we build upon previous work [23], which utilizes 3D convolutional layers to learn temporal domain features and adapt our network architecture such that it can capture the slow-changing temporal characteristics of the EDA signal.

## III. DATASET

### A. Recruiting and Recording

We recorded a dataset of $N = 20$ participants (5 female, 15 male, ages 19–36, $\mu = 26.7$ and $\sigma = 3.9$) to investigate remote sympathetic arousal prediction. Based on the Fitzpatrick scale [36], 5 participants had skin type II, 10 skin type III, 3 skin type V, and 2 skin type VI. The dataset captures 9.5 minutes of video recording (with disabled white balancing, autofocus, and auto-exposure) of participants' faces and hands, with synchronized EDA recordings from the finger and PPG recordings from the fingers and foreheads.

### B. Apparatus

The participants placed their heads on a chin rest and their hands on a table with their palms facing upwards (see Fig. 1). The hands were secured with a belt over the thumb to minimize any motion in the videos and we kept the lighting and temperature in the room constant throughout the study. We recorded the videos using two Basler acA1300-200uc cameras pointed toward the participants' faces and hands and the physiological signals from a synchronized BIOPAC MP160 that triggered the cameras by wire. The design of the study is based on that of previous work [20], as they have shown successfully that with their setup, it is possible to remotely measure sympathetic arousal. To the best of our knowledge, our dataset is the first for remote sympathetic arousal prediction that will be available to other researchers.

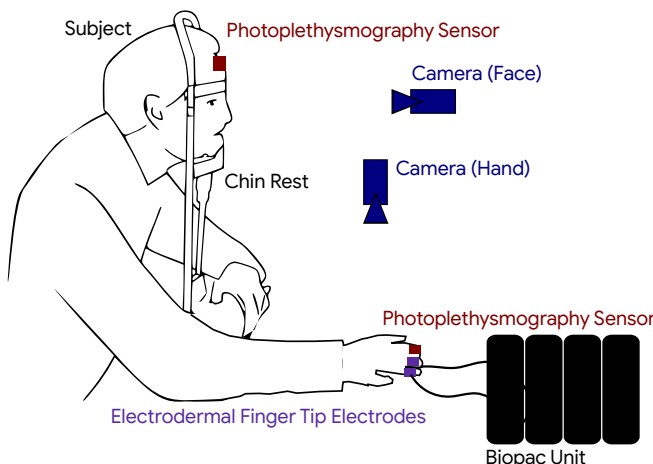

Fig. 1. The apparatus used for our study.

### C. Study Protocol

The protocol alternated between periods of resting (2 minutes) and periods of physical stress due to pain during which the participants pinched their skin (self-pinching, 30 seconds) to stimulate an EDA response, starting with a period of rest. To ensure that all participants had significant changes in EDA during the study, we used a dependent t-test for paired samples ($p > 0.05$), comparing the EDA levels during the periods of rest to the periods of stress. Two (10%) of the 20 participants did not have significant EDA responses during the study (consistent with previous work [20], [30]), and we did not consider them for the following evaluation.

## IV. METHODS

Our work comprises two main components. First, we aim to remotely estimate a person's sympathetic arousal corresponding to a contact-based EDA signal using only facial videos as input. We use a deep-learning-based approach as these approaches have shown stronger performance than signal-processing-based approaches for related problems such as rPPG prediction [22], [23], [33], [37]. Second, we train a classification model that uses our remotely predicted signals to detect if a person experiences physical stress due to pain.

### A. Remote Sympathetic Arousal Prediction

*1) Proposed Architecture:* As the backbone of our architecture, we use a 3D CNN [23] with a temporal input length of $T = 768$ frames. While such 3D CNN-based architectures have achieved impressive performance for the task of video-based HR prediction, most of the used 3D CNN architectures [23], [33] treat all input frames equally, ignoring that different frames may provide different contributions to the target prediction. For example, frames with more motion might convey less information than frames with less motion. To address this problem, we propose a temporal attention module (TAM) that allows our model to learn to discriminate between more and less important features along the temporal dimension. Each TAM block is composed of a 3D average pooling, a 3D convolutional layer (kernel size 1, stride 1,

padding 0), and finally, a multi-layer perceptron (MLP, using the ReLU activation function) with a reduction rate $r = 16$ followed by a sigmoid activation function (see Fig. 2). Given an image feature map $\mathbf{F_{in}} \in \mathbb{R}^{C \times T \times w \times h}$ as input, a TAM block infers a 1D temporal attention map $\mathbf{A_t} \in \mathbb{R}^T$, which is then broadcasted (copied) along the spatial and channel dimension during multiplication. The final output $\mathbf{F_{out}}$ of the attention process can be summarized as:

$$\mathbf{F_{out}} = \mathbf{A_t} \otimes \mathbf{F_{in}}, \tag{1}$$

where $\otimes$ denotes element-wise multiplication. We add one TAM block before each temporal up-sampling step. This approach is inspired by the success of sequentially using attention maps along the channel and spatial dimensions [38]. Fig. 2 shows the final architecture and the proposed TAM block. The total number of FLOPs for one batch using our model is about $100\,\mathrm{GigaFlops}$ and the total number of parameters of our model is about $790\,\mathrm{k}$. Of these, the two TAM blocks account for about $23\,\mathrm{k}$ parameters, which represents only about 3% of the total number of parameters.

*2) Implementation Details:* First, we detect the participants' faces using OpenCV's cascade classifier [39], crop the images to the bounding boxes, and then resize the images to a resolution of $72 \times 72$ (similarly as in previous work [20], [23]). Then, we downsample the frame rate of the videos from $100\,\mathrm{Hz}$ to $10\,\mathrm{Hz}$ as a frame rate of $10\,\mathrm{Hz}$ is sufficient for the typical frequency band of the sympathetic component of the EDA signal (between $0.045$–$0.25\,Hz$ [34]). Finally, we take the consecutive difference between the frames and standardize them by dividing them through the STD of the pixel intensity values [22]. We process the ground truth EDA signals in the same fashion and use them as labels for our model.

Afterward, we trained our model using leave-one-subject-out (LOSO) cross-validation, during which we iteratively held out the data of one participant as test set, one as validation set, and use the data of the remaining participants as training set. We used a batch size of 4 for 30 epochs, a learning rate of 0.001, and the mean squared error as the loss function. To validate the stability of the models, we report the mean obtained correlations across all participants and three random seeds, which helps to ensure that our experiments are reproducible and do not depend on a single random initialization, such as the weight initialization of the neural network. We trained our model on a GeForce RTX 4090 with a runtime of about 9 hours for all subjects and three random seeds.

*3) Evaluation:* To compare the similarity between our predicted and the ground truth signal, we use the Spearman correlation as proposed in previous work [20]. We evaluate the performance of our model to predict the raw EDA signal and the slower-acting tonic component of the EDA signal, which we obtain using the convex optimization approach [40]. As previous work has found that longer temporal inputs help to remotely predict sympathetic arousal [20], we also evaluate how different input window sizes ranging from $T = 256$ to $T = 1024$ frames (corresponding to 25.6 to 102.4 seconds) influence the performance of our network. In addition, we

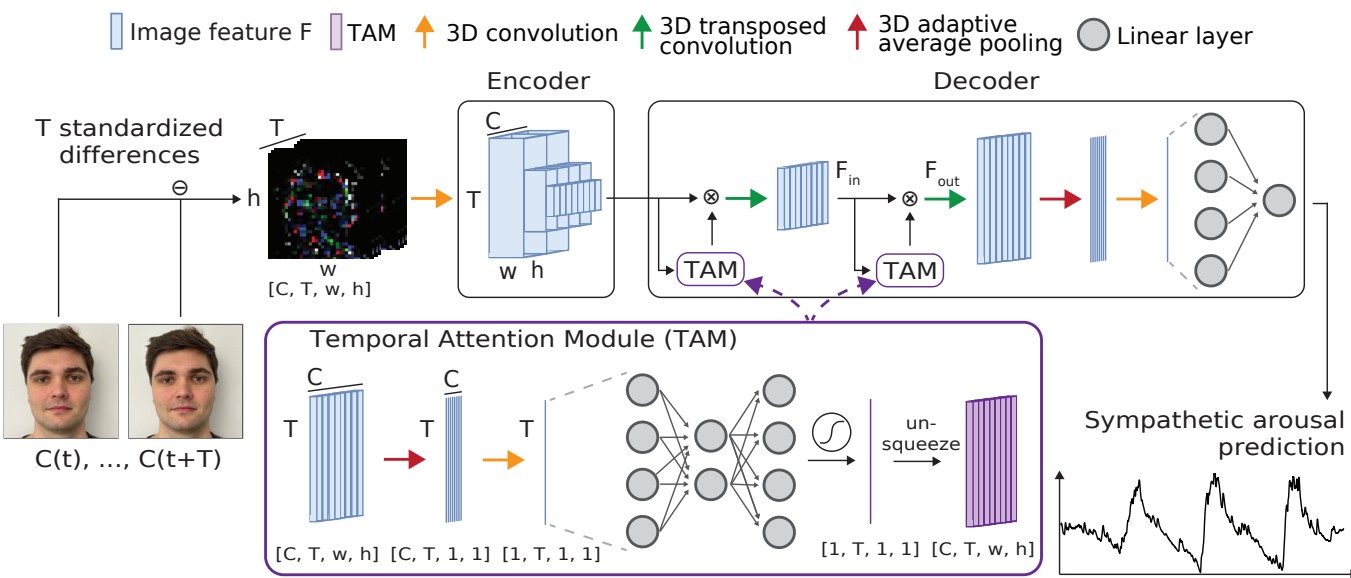

Fig. 2. Our proposed neural architecture to predict sympathetic arousal from facial videos.

implement four baseline networks (one Transformer-based model and three CNNs) that are used for remote HR prediction to compare the performance of our proposed model to the performance of these established networks. We trained all models in the same fashion as our proposed model. Furthermore, to compare our method with the current baseline, we also implemented the *blood pulsation amplitude* method (current baseline, a signal-processing-based approach) [20] and evaluated it on our dataset. A valid concern is that our model learns to predict small facial motions, such as micro-expressions, which could potentially occur during phases of stress. We, therefore, also calculate the Spearman correlation between our predicted signals and the magnitude of the optical flow of the face.

### B. Physical Stress Detection

To assess how much value our remotely predicted sympathetic arousal signal adds to the downstream task of physical stress detection due to pain, we perform a classification on the used dataset. The goal of the classification is to distinguish between the non-stressful periods (resting) and the 30 second stressful periods (self-pinching). We perform the classification separately using only the ground truth signals obtained from the contact measurement device (EDA and PPG) and the camera-based predicted signals (our remotely predicted sympathetic arousal trained with tonic EDA and a remotely predicted rPPG signal). To predict the rPPG signal from the camera, we use the original PhysNet network [23] trained on our dataset with a LOSO cross-validation approach. For both modalities, the contact-based and camera-based inputs, we each run the stress detection once using only the PPG/ rPPG signal, once using only the EDA/ remote sympathetic arousal signal, and once with both signals together. In this way, we aim to analyze the importance of the EDA/remote sympathetic arousal signal compared to the PPG/rPPG signal in this study setting. We divide the 9.5 minute recordings of each participant

TABLE I
THE CALCULATED STATISTICAL FEATURES FROM THE PPG/ RPPG AND THE CONTACT EDA/ PREDICTED SYMPATHETIC AROUSAL SIGNALS.

| Signal | Feature | Description |
|---|---|---|
| PPG/ rPPG | $\mu_{HR}$ | Mean HR |
| | $\min_{HR}$ | Minimum HR |
| | $\max_{HR}$ | Maximum HR |
| | $\mu_{HRChange}$ | Mean change of the HR |
| | SDNN | STD of NN intervals |
| EDA/ predicted sympathetic arousal | $\mu_{EDA}$ | Mean |
| | $\sigma_{EDA}$ | STD |
| | $\min_{EDA}$ | Minimum value |
| | $\max_{EDA}$ | Maximum value |
| | $\mu_{EDAChange}$ | Mean of consecutive change |

into 19 windows (3 windows of pinching and 16 windows of resting) of 30 seconds each without overlap. For each window, we extract 10 commonly used features for stress detection from the PPG/rPPG and EDA/sympathetic arousal signals, such as the mean HR or mean EDA (see Table I) [41]–[43]. As a classifier, we use the Gradient Boosting (GB) classifier. To train our classification algorithms, we again use LOSO cross-validation. Given the unbalanced number of stress and rest windows, we compute the balanced accuracy and F1-score to evaluate our model performance.

### V. RESULTS

### A. Remote Sympathetic Arousal Prediction

We show the mean correlation $\rho$ and the standard deviation of our results across all participants and three different random seeds in Table II. Using our proposed model with an input window size of 768 frames, we obtained a mean correlation of $0.73 \pm 0.01$ (raw EDA)/$0.77 \pm 0.02$ (tonic EDA) between our predicted signal and the ground truth EDA signal over all participants. This is an improvement of 40%/48% compared to using the current baseline method (Traditional Method [20],

a signal-processing-based approach) on our dataset. Also, the STD decreases using our method from 0.24/0.26 to 0.19/0.20.

We further evaluated the influence of the input window size on the model performance. The mean correlation gradually increases from a mean correlation of 0.37/0.47 using a window size of 256 frames to a mean correlation of 0.73/0.77 using a window size of 768 frames. When using a larger window size of 1024 frames, the mean correlation decreases to 0.65/0.71. The other implemented network structures, which are usually used for rPPG measurements, showed much lower performance than our introduced model, with mean correlations between 0.23 and 0.51. Furthermore, the Spearman correlation between the calculated magnitude of the dense optical flow of the facial video and our predicted signals is 0.23, indicating that our model does not simply learn to predict facial motions.

To qualitatively cross-check the results, we plotted our predicted sympathetic arousal and the ground truth EDA signals for all participants. In Fig. 3, we show four predicted signals and the corresponding ground truth signals. We can see that our predicted signals closely follow the overall trend of the ground truth signal. However, for individual participants, our model is currently only capable of accurately predicting the global trend and not smaller phasic changes, as we show in the bottom-right plot of Fig. 3.

### B. Physical Stress Detection

Table III summarizes the results of our physical stress (due to pain) classification using the GB classifier. A simple baseline classifier, which always predicts rest, would achieve a balanced accuracy (BACC) of 0.5 and an F1 score of 0.4. We obtain very similar maximum BACC and F1 scores for both modalities, the camera-based signals and the predicted camera-based signals. For both modalities, we obtain the highest BACC using only the features from the EDA/our remotely predicted sympathetic arousal signal. With only the contact-based signals, the highest BACC is 0.94, and the highest F1 score is 0.89. For the camera-based signals, the highest BACC is 0.90, and the highest F1 score is 0.83. However, for both the contact-based and camera-based signals, the BACC and F1 scores drop considerably when using only the PPG/rPPG signal compared to using the EDA/remotely predicted sympathetic arousal. For the contact-based signals, the BACC drops to 0.57 and the F1 score to 0.18 and for the camera-based signals, the BACC drops to 0.56 and the F1 score to 0.17. Using our remotely predicted sympathetic arousal improves the balanced accuracy by 61% compared to only using the remotely predicted rPPG signal.

## VI. DISCUSSION

### A. Remote Sympathetic Arousal Prediction

In our quantitative analysis, we have shown that we achieve a 40% (raw EDA)/48% (tonic EDA) higher mean correlation of 0.73/0.77 across all participants predicting the raw/tonic EDA signal using our introduced model compared to the current state-of-the-art method, which uses a signal processing approach [20]. At the same time, we decrease the STD of the

correlation across all participants from 0.24/0.26 to 0.19/0.20. Our qualitative analysis (see Fig. 3) also shows how closely our predicted sympathetic arousal follows the global trend of the ground truth EDA signal. Furthermore, we see a substantial performance improvement of our network using the TAM blocks compared to using other 3D CNN architectures like PhysNet [23]. This indicates that the TAM blocks help to learn the network to discriminate between more and less important features. Also, we evaluated the performance of our network using different window input sizes. The mean correlation gradually increases from a mean correlation of 0.37/0.47 using a window size of 256 frames (corresponding to 25.6 seconds) to a mean correlation of 0.73/0.77 using a window size of 768 frames (corresponding to 76.8 seconds) and then decreases again. This is consistent with previous work that obtained the best performance using a window size of 60 seconds [20]. In addition, the main spectral power density of an EDA signal lies in the frequency band between 0.045–0.25 Hz (corresponding to 4 to 22.2 seconds) [34], indicating that a window size of 22.2 seconds is beneficial to predict EDA. Our qualitative analysis shows that our obtained correlations of 0.73/0.77 indeed reflect that our predicted signals capture the overall trend of the ground-truth EDA signals accurately. However, we also recognize that our model is not yet capable of predicting smaller changes, and while our proposed model considerably improved the mean performance, a standard deviation of 0.19/0.20 still means that our model is not yet able to accurately predict sympathetic arousal for individual participants. Additionally, to show that our network does not learn to simply predict motions that could occur during phases of stress, we have calculated the correlation between our predicted signals and the magnitude of the dense optical flow of the videos of the participants' faces. We have obtained a mean correlation of only 0.23 across all participants, indicating that our network does not simply predict motion.

Finally, previous work suggests that we do not measure actual sweat responses when predicting sympathetic arousal from facial videos but changes in blood flow that correlate with sympathetic arousal (see Section II) [20]. Therefore, we expect that our measured signal from the face and the EDA signal do not perfectly match, e.g., due to small temporal offsets between the two signals. However, as we discuss below, we believe that our stress classification results show that for the downstream task of detecting physical stress, we do not need to be able to reconstruct the ground truth EDA signal perfectly.

### B. Physical Stress Detection

To help reveal where the predicted sympathetic arousal has utility for downstream inferences, we performed a physical stress (due to pain) detection experiment. Using camera-based predicted sympathetic arousal, we obtain a maximum BACC/F1 score of 0.90/ 0.83 for detecting physical stress due to pain with a GB classifier using our remotely predicted sympathetic arousal. This is almost as high as using the contact-based EDA signal with a maximum BACC/F1 score of 0.94/ 0.89. Furthermore, we see in Table III that using only

TABLE II

MEAN AND STD OF THE SPEARMAN CORRELATIONS ($\rho$) OVER THREE RANDOM SEEDS BETWEEN THE PREDICTED SYMPATHETIC AROUSAL AND THE GROUND TRUTH EDA SIGNALS ACROSS ALL PARTICIPANTS. OUR METHOD IMPROVES THE MEAN PERFORMANCE BY 40% (RAW EDA)/48% (TONIC EDA) TO 0.73/0.77 AND DECREASES THE STD BY 0.05/0.06 COMPARED TO THE TRADITIONAL METHOD (CURRENT BASELINE) [20].

| Method | Window | Raw | | Tonic | |
|---|---|---|---|---|---|
| | | Mean $\rho$ | STD | Mean $\rho$ | STD |
| TS-CAN [33] | 768 | $0.23 \pm 0.03$ | $0.32 \pm 0.02$ | $0.33 \pm 0.05$ | $0.33 \pm 0.04$ |
| PhysFormer [37] | 768 | $0.28 \pm 0.02$ | $0.24 \pm 0.02$ | $0.23 \pm 0.02$ | $0.28 \pm 0.03$ |
| DeepPhys [22] | 768 | $0.31 \pm 0.05$ | $0.26 \pm 0.02$ | $0.39 \pm 0.06$ | $0.28 \pm 0.03$ |
| PhysNet [23] | 768 | $0.43 \pm 0.02$ | $0.25 \pm 0.03$ | $0.51 \pm 0.01$ | $0.31 \pm 0.01$ |
| Traditional Method [20] | — | 0.52 | 0.24 | 0.52 | 0.26 |
| Ours | 256 | $0.37 \pm 0.06$ | $0.22 \pm 0.04$ | $0.47 \pm 0.03$ | $0.21 \pm 0.04$ |
| Ours | 384 | $0.49 \pm 0.04$ | $0.25 \pm 0.00$ | $0.58 \pm 0.03$ | $0.26 \pm 0.02$ |
| Ours | 512 | $0.64 \pm 0.00$ | $0.23 \pm 0.02$ | $0.63 \pm 0.00$ | $0.29 \pm 0.03$ |
| *Ours* | **768** | $\mathbf{0.73 \pm 0.01}$ | $\mathbf{0.19 \pm 0.01}$ | $\mathbf{0.77 \pm 0.02}$ | $\mathbf{0.20 \pm 0.01}$ |
| Ours | 1024 | $0.65 \pm 0.00$ | $0.28 \pm 0.02$ | $0.71 \pm 0.02$ | $0.29 \pm 0.01$ |
| Improvement of ours over best previous method | | **+0.21** | **−0.05** | **+0.25** | **−0.06** |

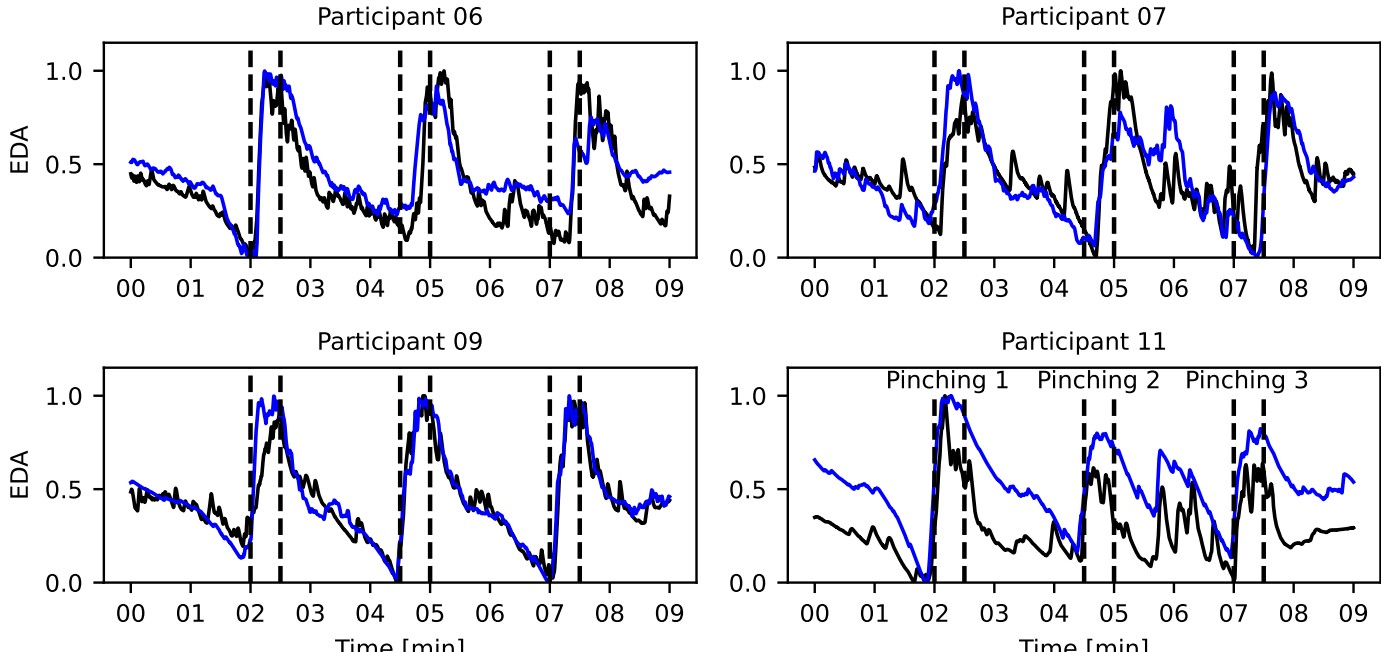

Fig. 3. Visual comparison between our predicted sympathetic arousal (blue) and the ground truth tonic EDA signal (black) for four participants. At minutes 2, 4.5, and 7, the participants are instructed to pinch themselves for 30 seconds to cause a sympathetic stress response. Note that for individual participants, such as participant 11, the model is currently only able to predict the global trend accurately. This difference is attributed to the nature of our method, which estimates sympathetic arousal by analyzing blood flow changes rather than measuring absolute EDA values.

either the contact-based PPG signals or only the remotely predicted rPPG signals, an accurate physical stress (due to pain) prediction is not possible for our dataset. A relatively modest stimulus like self-pinching does not seem to change features related to the blood volume pulse (BVP), such as heart rate, enough to allow for an accurate physical stress prediction. Very similar results were reported in related works. With contact-based signals, accuracies between 0.51 and 0.75 were obtained using only heart rate features (compared to 0.57 for our dataset) and accuracies of up to 0.95 with multi-modal features using the heart rate and the EDA signal to detect pain (compared to 0.94 for our dataset) [41], [44]. Previous work also obtained similar results using only features obtained

from a remotely predicted rPPG signal, achieving accuracies between 0.59 and 0.63 (compared to 0.56 for our dataset) [45], [46]. This highlights the limitations of relying solely on BVP and the importance of having a second physiological metric, such as our proposed remote sympathetic arousal, to detect physical stress.

### C. Limitations and Future Work

We recognize that our model and dataset have certain limitations. First, our study is highly controlled. To minimize any motion or lighting changes, we placed the participants' heads on a chin rest and kept the lighting constant in the room. Therefore, only limited conclusions can be drawn about the

TABLE III
BALANCED ACCURACIES (BACC) AND F1 SCORES USING ONLY THE
CONTACT-BASED SIGNALS (PPG AND EDA), ONLY OUR CAMERA-BASED
PREDICTED SIGNALS (rPPG AND REMOTE SYMPATHETIC AROUSAL
(rSA)), AND BOTH TOGETHER. NOTE THAT WE CAN ONLY PREDICT
PHYSICAL STRESS ACCURATELY USING EDA/OUR PREDICTED rSA AND
NOT USING THE PPG/rPPG SIGNALS.

| | PPG/rPPG | EDA/rSA | BACC | F1 |
|---|---|---|---|---|
| Contact (reference measurement) | ✓ | ✗ | 0.57 | 0.18 |
| | ✗ | ✓ | **0.94** | **0.89** |
| | ✓ | ✓ | 0.93 | 0.88 |
| Camera (estimated) | ✓ | ✗ | 0.56 | 0.17 |
| | ✗ | ✓ | **0.90** | **0.83** |
| | ✓ | ✓ | 0.88 | 0.80 |
| Baseline | — | — | 0.50 | 0.40 |

generalizability of our approach to more real-world situations. However, we believe that it is essential to first establish a dataset that makes it possible to evaluate the feasibility of possible methods under more controlled conditions. In future work, we aim to extend our dataset to include more natural scenarios and to evaluate our model's robustness to such conditions. Second, while we showed in our qualitative analysis that our model can predict the global trend of the EDA signal, we also found that it is not yet capable of predicting the small phasic components. We believe that one promising approach could be to investigate different loss functions, which give greater weight to the errors of the phasic component. This could help to improve the model's sensitivity to the more rapid phasic fluctuations. Finally, our dataset comprises data from 5 female and 15 male participants. We acknowledge that we should have paid more attention to a balanced ratio to create a balanced dataset and to potentially also allow for an investigation of the performance differences of our approach for female and male participants. We aim to correct this oversight in the future by recording further participants.

### D. Broader Impacts

Perhaps the most obvious application for measuring sympathetic arousal remotely is stress management. Previous work has shown that EDA/sympathetic arousal measurements can be used to help people better understand their stress patterns [47]. In addition, currently used wearable sensors, such as smartwatches, can be very inconvenient for the user, or it might not be possible to wear them for safety reasons, e.g., for assembly line workers. Using a camera in such cases could offer unique opportunities for deployment, otherwise hard to achieve. Finally, we believe that it is important to consider the potential for a new technology such as ours to be deployed with negligence or by a bad actor. While people might be able to hide their emotions by not expressing them, they are, in general, not able to control their physiological states. Therefore, it is important that there are mechanisms in place to be aware of remote measurements and consent to them.

## VII. CONCLUSION

In this paper, we have demonstrated that it is possible to improve the performance of remote sympathetic arousal prediction from a video of a person's face by using a 3D CNN architecture tailored to the temporal dynamics of sympathetic arousal. To evaluate our approach, we contribute the first dataset specifically designed for the task of remotely predicting sympathetic arousal and make it available to other researchers. Using LOSO cross-validation, we have demonstrated that our proposed network obtains a mean correlation of up to 0.73 (raw EDA)/0.77 (tonic EDA) between the predicted sympathetic arousal and the ground truth EDA signal, marking a 40%/48% improvement compared to previous work. However, we also recognize that our model is not yet capable of predicting more detailed phasic changes of the EDA signal for individual participants. Furthermore, we trained a GB classifier with features extracted from the contact EDA and PPG signals and our remotely predicted sympathetic arousal and rPPG signals to detect physical stress caused by pain. We achieved a mean BACC of 0.90 in predicting physical stress using only our remotely predicted signals. Our stress classification experiments also revealed that using contact PPG and remotely predicted rPPG signals alone does not yield accurate results for physical stress detection due to pain in our dataset. This underlines the importance of our proposed approach to offer an alternative physiological measurement to accurately predict physical stress. We hope that our contributed network and dataset can assist other researchers in exploring various signal processing and machine learning techniques for developing accurate remote sympathetic arousal prediction models.

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
