# OpenReview forum: "SympCam: Remote Optical Measurement of Sympathetic Arousal"
_IEEE.org/EMBS/BHI/2024/Conference — IEEE BHI'24_

### Official Review · Reviewer_A2o2 · 2024-08-14
**Review of SympCam: Remote Optical Measurement of Sympathetic Arousal**

**Overall Rating:** 7
**Confidence:** 4

**Other Quality Metrics:**

Clarity of Writing: Great
Clinical Significance: Excellent
Methodological Novelty: Great
Experiments and Results: Great

**Questions For The Authors:**

Could the model be adapted or fine-tuned to better capture the more minor phasic changes in the EDA signal?

**Strengths:**

Using a 3D convolutional network with TAM to predict sympathetic arousal is innovative and well-implemented. The model significantly outperforms existing methods, improving prediction accuracy by 48%. Creating a dataset designed explicitly for this task is valuable for the community, providing a standardized benchmark for future research. This work has significant implications for telehealth and stress management, offering a non-invasive and scalable solution.

**Summary Of The Paper:**

The paper presents SympCam, a deep learning model that utilizes a 3D convolutional neural network with a Temporal Attention Module (TAM) to forecast sympathetic arousal from facial videos. The model's predictions strongly correlate with electrodermal activity (EDA), a widely accepted measure of sympathetic arousal. The method proposed in the paper is compared to existing approaches, emphasizing significant improvements in accuracy. Furthermore, a new dataset designed explicitly for remote sympathetic arousal prediction will be available to the research community.

**Weaknesses:**

The study was conducted in a highly controlled environment, potentially limiting the model's applicability in real-world settings. Despite capturing overall trends, the model struggles to predict minor, phasic changes in the EDA signal.

---

### Official Review · Reviewer_Kj3u · 2024-08-16
**Good work**

**Overall Rating:** 7
**Confidence:** 4

**Other Quality Metrics:**

(a) Clarity of writing; Excellent.
(b) Clinical Significance; Great
(c) Methodological Novelty; Great.
(d) Experiments and Results. Excellent.

**Questions For The Authors:**

1. As the author discussed, the dataset is very limited and only collected data from a constant environment. However, It's better to collect data on different settings like different light conditions or different face angles.  To make it more practical. Because the fixed angle and ideal condition are not always available in telehealth monitoring applications.

2. In Sec.IV,A,2. You measure correlations across all participants and three random seeds. what is "three random seeds"? Please provide more explanation to make it clear.

3. In Fig.3. For participant 11, the prediction can't follow the ground truth after the first drop after 2.5 minutes. However, it follows very well on all other three participants. Could you please explain what causes this difference?

**Strengths:**

The paper is well-organized and easy to follow. The evaluation is very comprehensive.

**Summary Of The Paper:**

The author proposed the first facial video-based AI sympathetic arousal monitoring system to facilitate telehealth and stress monitoring. The dedicated dataset is collected. The final performance reveals the effectiveness and efficiency of the proposed method.

**Weaknesses:**

The model and dataset have certain limitations as the author also discussed in the paper. Explanation of several points is needed to eliminate potential misunderstanding.

---

### Official Review · Reviewer_uY2s · 2024-08-27
**Review on SympCam: Remote Optical Measurement of Sympathetic Arousal**

**Overall Rating:** 7
**Confidence:** 3

**Other Quality Metrics:**

(a)Clarity of writing: fair
(b)Clinical Significance: good
(C)Methodological Novelty: fair
(d) Experiments and Results:f air

**Questions For The Authors:**

The study is very controlled, especially the set-up. Which makes the proposed method unusable in real-life applications. What will be done differently to make the proposed methods usable in real-world scenarios?

**Strengths:**

The manuscript provides a thorough review of existing literature, clearly identifying the limitations of previously published works and differentiating their approach from past research. The authors recruited 20 participants and utilized a LEAVE ONE SUBJECT OUT (LOSO) cross-validation method to ensure robust validation of their model.

The authors used a 3D CNN as the backbone of their architecture, acknowledging the strengths of 3D CNNs in video-based heart rate (HR) prediction. However, they identified a key limitation: traditional 3D CNN architectures typically treat all input frames equally, overlooking the varying contributions that different frames may have on the target prediction. To address this, the authors proposed a temporal attention module (TAM), enabling their model to distinguish between more and less significant features along the temporal dimension.

The manuscript focuses on two main components: Remote Sympathetic Arousal Prediction and Physical Stress Detection. The authors have also contributed a novel dataset specifically designed for the task of remotely predicting electrodermal activity (EDA). This dataset is made available to other researchers upon request, offering a valuable resource for further advancements in this field.

**Summary Of The Paper:**

This manuscript presents SympCam, a novel 3D convolutional architecture designed for remote sympathetic arousal prediction. The authors enhance the model's sequential data processing capabilities by incorporating a temporal attention module (TAM), which improves temporal coherence. Their method shows a significant improvement, boosting accuracy metrics of sympathetic arousal prediction by 48% compared to previous work, achieving a mean correlation of 0.77. The manuscript also provides a comparative analysis with existing remote photoplethysmography (rPPG) networks, demonstrating that these networks alone cannot accurately predict sympathetic arousal "out-of-the-box." The authors further highlight that their method achieves a balanced accuracy of 90% in detecting physical stress, representing a 61% improvement over traditional rPPG methods. Additionally, the manuscript introduces a new dataset specifically designed for remote sympathetic arousal prediction, which is a valuable contribution to the field.

Overall, the study is well-structured and provides strong evidence for the effectiveness of SympCam in addressing the challenges of remote sympathetic arousal prediction. The integration of TAM and the comparative analysis with rPPG methods underscore the manuscript's innovative approach. However, further details on the dataset and broader validation across diverse scenarios could strengthen the manuscript.

**Weaknesses:**

Novelty:
Authors mentioned that ‘Building on that work [21] and the correlation between blood flow and EDA on the forehead [31], [32], subsequent work has shown that sympathetic arousal can also be inferred from videos of the face using only a regular RGB camera by measuring the peripheral blood flow to the forehead [22]. While this work showed first proof that it is possible to remotely extract a person’s sympathetic arousal from a video of the face, the mean correlations across participants were moderate, and the standard deviation (STD) was high. Nevertheless, we were inspired by these results and aimed to develop a more robust method and release a dataset for remote sympathetic arousal prediction that is also available to other researchers.’
This means that the work is not novel rather improved than previously published literature.
Authors also mentioned the limitations of the study as following:
“We recognize that our model and dataset have certain limitations. First, our study is highly controlled. To minimize any motion or lighting changes, we placed the participants’ heads on a chin rest and kept the lighting constant in the room. Therefore, only limited conclusions can be drawn about the generalizability of our approach to more real-world situations. Second, while we showed in our qualitative analysis that our model can predict the global trend of the ground truth signal, we also found that it is not yet capable of predicting the small phasic components. Finally, our dataset comprises data from 5 female and 15 male participants. We acknowledge that we should have paid more attention to a balanced ratio to create a balanced dataset and to potentially also allow for an investigation of the performance differences of our approach for female and male participants. We aim to correct this oversight in the future by recording further participants.”

But authors did not mention how they will approach to solve these limitations. For example the study setup seems very controlled and not usable in real-life. What can be done differently to apply these methods in real-life?

---

> ### Author Rebuttal · Authors · 2024-08-30
>
> We would like to thank the reviewer for the feedback and for taking the time to review our paper. We are pleased to hear that the reviewer finds that our study provides strong evidence for the effectiveness of SympCam and that our manuscript's approach is innovative. We will address the two raised concerns and questions below and have marked the corresponding changes in the revised version of our paper in blue.
>
>
> **1) Novelty.** \
> We fully acknowledge that the problem of remote sympathetic arousal prediction was first introduced by Braun et al. [1]. As noted by the reviewer, we have explained this in Sections II and III of our paper, and it was *not* our intention to suggest otherwise. To clarify this, we have explicitly mentioned this in Section I. of the revised version of our paper.\
> The novelty of our work lies not in the initial introduction of this problem, but rather in our key contributions: 1) we introduce the first neural model specifically developed for remote sympathetical arousal prediction, achieving a 48% increase in accuracy compared to prior approaches; 2) we contribute the first publicly available dataset designed for this problem, which we believe will be a valuable resource for the research community; and 3) we demonstrate the practical value of remote sympathetical arousal prediction for physical stress detection.
>
> **2) Controlled Environment and Real-World Application.** \
> We acknowledge that the controlled nature of our current study setup may limit its direct applicability to real-life scenarios. Please see our general response below for our explanation of why we designed our dataset this way. However, we cannot yet evaluate whether our model will also work effectively in more natural scenarios without such a dataset. Once we have extended our dataset to include more real-world applications, we will assess the model's performance under these conditions. If necessary, we would then implement, e.g., data augmentation and domain adaptation techniques to enhance the model's robustness against variations such as lighting, head movement, and other real-world factors. We have included this aspect in the discussion in the final version of our paper in Section VI. C.
>
>
> We hope that we were able to address the reviewer’s questions regarding the novelty of our paper and the applicability of our method in real-life settings. If anything remains unclear or if further questions arise, we would be very happy to address them.
>
> [1] B. Braun, D. McDuff, T. Baltrusaitis, and C. Holz, “Video-based sympathetic arousal assessment via peripheral blood flow estimation,” Biomedical Optics Express, vol. 14, no. 12, pp. 6607–6628, 2023.

---

### Decision · Program_Chairs · 2024-09-23

Accept